# Development and Deployment of a Framework to Prioritize Environmental Contamination Issues

**Nicholas D. Kim** [1], **Matthew D. Taylor** [2], **Jonathan Caldwell** [2], **Andrew Rumsby** [3],
**Olivier Champeau** [4] **and Louis A. Tremblay** [4,5,*]

1   School of Health Sciences, Massey University Wellington, P.O. Box 756, Wellington 6140, New Zealand;
    n.kim@massey.ac.nz
2   Waikato Regional Council, P.O. Box 4010, Hamilton East, Hamilton 3247, New Zealand;
    matthew.taylor@waikatoregion.govt.nz (M.D.T.); jonathan.caldwell@waikatoregion.govt.nz (J.C.)
3   EHS Support, PO Box 15887, New Lynn, Auckland 0640, New Zealand; andrew.rumsby@ehs-support.com
4   Cawthron Institute, Private Bag 2, Nelson 7042, New Zealand; olivier.champeau@cawthron.org.nz
5   School of Biological Sciences, University of Auckland, P.O. Box 92019, Auckland 1142, New Zealand
*   Correspondence: louis.tremblay@cawthron.org.nz; Tel.: +64-3-539-3290

**Abstract:** Management and regulatory agencies face a wide range of environmental issues globally. The challenge is to identify and select the issues to assist the allocation of research and policy resources to achieve maximum environmental gain. A framework was developed to prioritize environmental contamination issues in a sustainable management policy context using a nine-factor ranking model to rank the significance of diffuse sources of stressors. It focuses on contamination issues that involve large geographic scales (e.g., all pastoral soils), significant population exposures (e.g., urban air quality), and multiple outputs from same source on receiving environmental compartments comprising air, surface water, groundwater, and sediment. Factor scores are allocated using a scoring scale and weighted following defined rules. Results are ranked enabling the rational comparison of dissimilar and complex issues. Advantages of this model include flexibility, transparency, ability to prioritize new issues as they arise, and ability to identify which issues are comparatively trivial and which present a more serious challenge to sustainability policy goals. This model integrates well as a planning tool and has been used to inform regional policy development.

**Keywords:** diffuse contamination; environmental management; priority ranking; local government; air pollution; stormwater; anthropogenic stressors; agricultural runoff; New Zealand; geothermal

## 1. Introduction

Regulatory agencies around the world face a wide range of environmental contamination issues. Territorial agencies must choose between issues when deciding where research and policy resources will be allocated with so many chemicals entering our natural environment [1]. Resources will always be limited. Ideally, they should be allocated in proportion to the seriousness of each issue in relation to policy goals. A 'contaminant' is defined as something that is out of place in an ecosystem, either because it does not occur naturally, or because it is present at significantly higher than natural background levels. A contaminant is regarded as a 'pollutant' when its concentration has become high enough to interfere with natural processes to produce one or more undesirable effects. Pollution of air, soil, water, or sediments can cause a wide range of sustainability problems. For example, accumulation of a trace metal in agricultural soils may cause one or more of the following: direct toxicity (to microbes, invertebrates, plants, grazing animals or wildlife), changes to soil chemistry, contamination of local ground water, non-compliance with food standards, increased dietary intakes in humans, accumulation in nearby river and lake bed sediments, and toxicity to aquatic ecosystems.

There are multiple sources of contaminants. Point source industrial discharges are often subject to site-specific controls which have led to major improvements in the quality of waste discharges. However, there has been less progress in managing multi-source or wide-scale contamination diffuse pollution sources. Diffuse pollution has been defined as: 'pollution arising from land use activities (urban and rural) that are dispersed across a catchment or sub catchment, and do not arise as a process industrial effluent, municipal sewage effluent, deep mine or farm effluent discharge' [2]. Although useful as a guide, the exclusions in this definition mean that it does not match all contexts: for example, in New Zealand, farm effluent discharges are recognized as a major source of diffuse pollution. Urban stormwater is widely recognized as a major diffuse source of pollutants released in the urban environment that contribute to the deterioration of receiving water bodies [3]. The intensification of agricultural activities results in another important diffuse source of anthropogenic contaminants in agricultural soils due to the use of significant quantities of fertilizers and pesticides to optimize production and achieve food security targets [4,5]. Atmospheric deposition is another key source facilitating the transport and dispersion of pollutants and materials in the atmosphere either through precipitation or deposition in dry weather [3,6,7].

All sources of non-point and widespread pollution are complex problems and constitute a major challenge for environmental regulators and policymakers globally [8]. The management of diffuse contamination is challenging as it results in the discharge of mixtures of multiple compounds in environmental media [9]. One of the major challenges of assessing the risk of complex environmental mixtures is the identification of those chemicals that contribute significantly to observed effects. Methods such as effect-directed analysis can assist to establish the corresponding cause–effect relationships and provide focus for potential management measures [10,11].

This study presents the development and validation of a framework to prioritize 117 environmental contamination issues in the Waikato region of New Zealand from a sustainable management policy context. The focus is on contamination issues that involve large geographic scales (e.g., all pastoral soils), significant population exposures (e.g., urban air quality), or multiple instances of the same source (e.g., 8000 former sheep-dip sites). Such issues are not managed under specific discharge consent provisions but may be subject to current or future regional planning rules. Outcomes of the results are discussed in relation to their implementation by staff of the relevant government organization for environmental protection (Waikato Regional Council) to decide research priorities and inform policy development.

## 2. Materials and Methods

### 2.1. Study Area

The Waikato region covers a 25,000 km$^2$ area and is predominantly rural with significant natural geothermal areas with a population over 458,000. Dominant rural activities and their approximate land areas are pastoral farming (dairy, beef, and sheep: ~1,430,000 ha), plantation forestry (~330,000 ha), and horticulture (~10,000 ha over the top 10 crops) [12].

### 2.2. Development of the Framework

#### 2.2.1. Identification of Contamination Issues

Although many disparate environmental contamination issues are known, it is common for each these to be considered individually and on a case-by-case basis, often only after a trigger has been reached. As a first step, any systematic attempt to prioritize environmental contamination issues requires the opposite approach: a rational method to identify and capture known and potential contamination issues in a single list, irrespective of external interests or preconceptions. The first step was to develop a broad and 'reasonably comprehensive' inventory of contamination issues for the study area (New Zealand's Waikato region). Then source categories were linked with the main

environmental compartments where substances can accumulate including air, soil, groundwater, surface water, and sediment. A list of natural (26) and anthropogenic (38) sources was compiled that may result in contamination of different environmental compartments (Table 1).

This 'source category and discharge'-based approach was taken on the basis that all types of environmental contamination require a source, and most sources involve a discharge—i.e., a transfer of contaminants from the source to one or more environmental compartments. Each of the 64 source categories was assigned a two-letter code, number, and short description as shown in Table 1. The first letter in these codes denotes the recipient environmental compartment (A = air, S = soil, G = groundwater, W = surface water and bed sediments) the second letter (N or A) whether the contamination is of natural or anthropogenic origin, and the number and description distinguish between different sources within a given category. For many contamination issues, significant movement occurs between environmental compartments, e.g., a contaminant discharged to air may subsequently deposit on soil. These links were recorded in issue descriptions.

Use of this source-list enabled a systematic approach to developing an inventory, through consideration of known or suspected contamination issues applicable to each source category in turn. Environmental chemistry expertise and familiarity with the study area was required for this step, with some judgments made about which issues were trivial enough to be safely excluded at the early stage. The resulting list comprised 117 known and potential environmental contamination issues of possible significance, with each one being identified by its source category code (Table 1), an additional numeral and a short description. To avoid duplication, this is not shown as a single list because the issues identified are all shown in the ranked results presented for each environmental compartment (Table 2, Tables S2 and S3).

Some source categories had no known significant instances of contamination linked to them, whereas others had several. The number of issues identified for each compartment were air: 28, soil: 44, groundwater: 23, and surface waters and bed sediments: 29. Although this inventory was developed for the Waikato region of New Zealand, most of the potential or known contamination issues would also apply to other predominantly rural regions around the world. Others occur in specific areas around the world where the sources are prominent, for example potential for arsenic and mercury contamination is commonly associated with natural geothermal systems as seen in Iceland, Japan and parts of the US.

The inventory developed in this work should not be viewed as comprising all known and potential contamination issues within the study region. Rather, as new concerns are identified, they can readily be added under an appropriate source code and ranked to determine their relative priority within the larger set, as discussed below. This approach would be preferable to a common situation that occurs, where newly identified concerns can take on an artificial importance, causing unnecessary diversion of institutional resources away from more significant resource management problems.

A structurally and philosophically unique feature of this model (compared with other risk assessment methodologies) is the ground-up baseline assumption that every instance of diffuse contamination to any environmental compartment is caused by a discharge, or source, of the contaminant to that compartment; and that such source may be either natural or anthropogenic. Creating a baseline framework using these concepts (Table 1) provides for a comprehensive coverage of all broad source categories without initially having to know the identities and natures of every discharge. This approach has the benefits of forcing expert consideration of discharges that may exist within each broad category, leading to early inclusion of potentially important sources that would normally be omitted; providing for natural as well as anthropogenic contamination sources; and making it easy to add new issues within the pre-existing framework as they are identified or arise.

**Table 1.** Natural and anthropogenic source categories of substances in air, soil, groundwater, and surface waters and their bed sediments.

| Environmental Compartment. | Natural Source Code and Category | Anthropogenic Source Code and Category |
|---|---|---|
| Air | **AN1** Synthesis in the atmosphere<br>**AN2** Entrainment of crustal material<br>**AN3** Entrainment of oceanic salts<br>**AN4** Biogenic emissions from living plants<br>**AN5** Dispersion of native plant pollens<br>**AN6** Volatilization from soils<br>**AN7** Volcanoes and geothermal areas<br>**AN8** Wild forest fires<br>**AN9** Global distillation<br>**AN10** Cosmic dust<br>**AN11** Gases generated through biological processes | **AA1** Industrial point-sources<br>**AA2** Fossil-fuel combustion<br>**AA3** Solid-fuel combustion<br>**AA4** Pollens from pastoral grasses, plantation forestry, urban trees, etc.<br>**AA5** Aerial topdressing or spreading of fertilizers<br>**AA6** Increased entrainment of crustal material after land clearance<br>**AA7** Use of pesticide sprays or fumigants<br>**AA8** Natural substances associated with farming of animals<br>**AA9** Weapons testing or use<br>**AA10** Long-range pollutant transport<br>**AA11** Waste incineration<br>**AA12** Greenhouse gases released through combination of sources AA1, AA2, AA3, AA7, AA8 and AA11<br>**AA13** Smoking<br>**AA14** Indoor sources |
| Soil | **SN1** Weathering of parent rocks and minerals<br>**SN2** Wet or dry deposition, preceded by any of the natural sources to air<br>**SN3** Sorption from groundwater or geothermal springs<br>**SN4** Concentration through biogenic or physical processes<br>**SN5** In situ generation through microbial or abiotic processes<br>**SN6** Degradation of complex organic material | **SA1** Anthropogenic discharge to air followed by wet or dry deposition<br>**SA2** Use of soil treatments on land<br>**SA3** Use of pesticides (herbicides, insecticides, fungicides)<br>**SA4** Use of plant supplements, veterinary medicines, or animal remedies<br>**SA5** Excretion of natural substances from farmed animals<br>**SA6** Sorption from irrigation water<br>**SA7** Loss of natural substances from exotic plants<br>**SA8** Fixation from air or water by exotic species<br>**SA9** Deposition to soil at industrial sites through spills, storage, local air discharge, or inappropriate disposal<br>**SA10** Weathering, ablation or renovation of human artifacts<br>**SA11** Release associated with mining activities or removal of overburden<br>**SA12** Creation of landfills, monofills or hazardous waste repositories. |

**Table 1.** *Cont.*

| | | |
|---|---|---|
| **Groundwater** | **GN1** Geothermal discharges into groundwater<br><br>**GN2** Leaching, preceded by any of the natural sources to soil | **GA1** Discharge to groundwater, preceded by any of the anthropogenic sources to soil<br>**GA2** Alteration of the groundwater environment<br>**GA3** Direct discharge into ground<br>**GA4** Fracking and underground gasification of coal |
| **Surface waters and bed sediments** | **WN1** Losses from stream beds and banks<br>**WN2** Surface runoff from land<br><br>**WN3** Direct inputs of groundwater<br><br>**WN4** Geothermal systems<br><br>**WN5** Wet or dry deposition, preceded by any of the natural sources to air<br>**WN6** In situ generation through microbial or abiotic processes<br>**WN7** Degradation of complex organic material | **WA1** Losses from stream beds and banks<br>**WA2** Anthropogenic discharge to air followed by wet or dry deposition<br>**WA3** Inputs of contaminated surface runoff, preceded by any of the anthropogenic sources to soil<br>**WA4** Inputs of contaminated groundwater, preceded by any of the anthropogenic sources to groundwater<br>**WA5** Direct discharge to water from industrial point-sources<br>**WA6** Urban storm water runoff<br>**WA7** Altered physical or biogeochemical processes<br>**WA8** Discharge to water through spills, weathering, ablation or renovation of human artifacts |

**Table 2.** Basic structure of the nine-factor model used to score and rank the significance of diffuse contamination issues.

| Factor | Scoring Scale | Factor Weighting | Highest Possible Score |
|---|---|---|---|
| 1. Scale—geographic or size of exposed population | 1 to 5 | 2 | 10 |
| 2. Accumulation capacity | 1 to 15 | 1 | 15 |
| 3. Reversibility | 0 to 5 | 1 | 5 |
| 4. Human health—potential for chronic harm | 0 to 5 | 3 | 15 |
| 5. Human health—potential for serious acute harm | 0, 2.5, or 5 | 1 | 5 |
| 6. Environmental impact | 0 to 5 | 2 | 10 |
| 7. Impact on animal welfare and production | 0 to 5 | 2 | 10 |
| 8. Harm to trade | 0 to 5 | 1 | 5 |
| 9. Reduction in land use flexibility | 0 to 5 | 1 | 5 |

Identification of the individual discharge types within each source category was undertaken by expert practitioners with expertise in air, soil, surface and groundwater quality monitoring, environmental chemistry, chemical contamination, resource allocation, and regulatory compliance. The systematic approach taken was to (a) record instances of diffuse contamination within each source category (Table 1) that were already known or were suspected, (b) identify additional possibilities by examining the extent to which each of the major natural and anthropogenic source categories (Table 1) are operative within the biogeochemical, physical and resource use contexts set by the Waikato region (e.g., "natural generation of the greenhouse gases $NH_4$ and $N_2O$ from wetlands" has ceased to be a diffuse contamination issue as nearly all wetlands in the Waikato region are impacted by drainage, but this drainage of wetlands has created a new issue "Release of $CO_2$ from drained wetlands"), and (c) exclude potential sources that were so localized, transient (e.g., the appearance of $^{109}Cd$ in New Zealand rainwater from Pacific Islands atomic weapons testing) or minor that it is very unlikely they would be capable of causing diffuse contamination in the first place, or inducing more than minor adverse effects.

2.2.2. Ranking Approach

Environmental contamination issues vary widely in their reach and consequences. However, in a sustainability context it is reasonable to define the most serious issues as those of the largest scale, which grow worse over time, exert deleterious effects (to humans, ecosystems, productive capacity, an economy, or resource capacity) and are irreversible. The ranking model of this framework is based on scores and weightings given to nine factors which capture these ideas. The basic structure of the ranking model is summarized in Table 2 and Table S1 and outlined further below.

In Table 2, the 'Scoring scale' was the only component that was ranked. For each factor and issue, a score was independently assigned, by each of four experts. The recommended approach was for each expert to assign scores to one factor at a time (across all issues), then sort those from highest-to-lowest to verify that they were comfortable with how each issue scored relative to the others. For each issue, the arithmetic average of each score from the four experts was then used in subsequent calculations. It was found that in general, scores assigned by different experts resulted in the same (or very similar) high-to-low placements, despite differences in the absolute values assigned. This gave some confidence in the reproducibility of the ultimate rankings.

- Factor 1. Scale

Scale can be considered in two ways: geographic area, and for human exposures, the population exposed. Scale of an issue was allocated a score of 1 to 5 based on calibration points shown in Table S1. Scores for scale were given a factor weighting of 2, providing a maximum possible score of 10 (Table 1).

- Factor 2. Accumulation capacity

There are three ways by which substances, or the harm they could cause, may accumulate: (a) the substance accumulates in the environmental compartment of interest over time, (b) the substance pools in an area from multiple sources, or (c) potentially harmful effects of long-term exposure are cumulative. Accordingly, this factor was characterized using three sub-components (outlined below), each of which was allocated a score from 1 to 5.

For progressive accumulation within an environmental compartment to achieve a high score, three aspects had to be satisfied: the contaminant is routinely added, the majority added is retained, and the relative concentration increase is significant. Two such examples of progressive accumulation are fluorine accumulation in phosphate-fertilized soils, and carbon dioxide accumulation in the global atmosphere.

Accumulation caused by pooling of an otherwise transient contaminant occurs for some air and water discharges. Pooling in this context is where environmental conditions cause accumulation of a contaminant discharge which would ordinarily disperse. For example, air pollutants from wood burned for domestic home heating can pool as a result of calm conditions or a thermal inversion layer. Winter air pollution is a significant problem in cities and towns of the Waikato region, home heating can be responsible for up to 90% of total particulate matter less than 10 microns in diameter (PM10) and 91 percent of PM2.5 on winter days [13–15]. Vehicle emissions are also a source of hazardous air pollutants, but with pooling being more localized to areas around major traffic corridors.

A higher accumulation score was also applied if potentially harmful effects of long-term exposure would be cumulative. Some substances may accumulate in the body with age until they pass toxic thresholds, and others can cause incremental damage which progressively increase disease risk. Accumulation of damage can occur through exposure to carcinogens (e.g., indoor radon, arsenic in drinking water, asbestos) or neurotoxic agents (e.g., many pesticides, solvents). Scoring included consideration of bioaccumulation and biomagnification where these processes were thought to be contextually and toxicologically relevant.

These three components of accumulation are relatively independent, and scores were therefore added, which (with no extra weighting applied) gave a maximum possible score for Factor 2 of 15 (Table 2).

- Factor 3. Reversibility

Partnered with accumulation potential, reversibility denotes the capacity for an environmental compartment to return to its natural state once a source of contamination is removed. Contamination of urban air with polycyclic aromatic hydrocarbons from diesel fumes is reversible over time through various dilution processes. By contrast, anthropogenic carbon dioxide emissions are less reversible, because carbon dioxide has a long atmospheric residence time. Trace metal contamination of soils and lakebed sediments is persistent, whereas many xenobiotic organic compounds (e.g., most modern synthetic pesticides) degrade. Reversibility was scored from 0 to 5 with calibration points as shown in Table S1. Within this factor a high score of 5 would imply that even if sources were removed, the attained level of contamination would be maintained in that environmental compartment for a long time, e.g., years to decades. Conversely, at the other end of the scale a score of 0 implies that the contamination of that compartment is readily reversible once the source is removed.

- Factors 4 and 5. Potential harms to human health

Protection of human health is a universally important policy goal reflected in legislation, planning rules and case-law. Harm to human health can occur through two distinct mechanisms, and so was addressed in the model through use of two factors (Table 2 and Table S1).

Chronic harm (Factor 4) can occur through long-term exposure and is typically at a low level but often across many people. This factor which was scored from 0–5 depending on the likelihood of

exposure and nature of potential harm: from no impact to a likely impact on most exposed people (Table S1). Acute harm (Factor 5) occurs through short-term high dose exposures, and in environmental settings this possibility mainly applies to toxic gases. Exposure to high levels of hydrogen sulfide ($H_2S$) gas has been linked with 10 human fatalities in the New Zealand geothermal resort town of Rotorua since the 1930s, and accidental carbon monoxide poisoning is a routine cause of fatalities around the world. Unlike chronic harm, serious acute harm does not require the progressive accumulation or the accumulation of damage, although localized pooling may be involved. It is infrequent and may involve few people, but when it occurs its consequences can be severe for individuals affected. Instances of contamination with capacity to cause acute harm did not necessarily show high scores for other factors. Acute harm was therefore added as a separate factor (Factor 5) to ensure its potential significance was reflected in the final rankings. Potential for a contamination issue to cause acute harm was allocated a score of 0, 2.5 or 5 based on descriptions shown in Table S1. The possibility of fatality occurring was assessed based on toxicology and likely exposure pathways, within the regional context. With weightings applied (Table 2) the two human health factors (Factors 4 and 5) gave a maximum possible score of 20.

- Factor 6. Environmental impact

Factor 6 reflects potential for the contamination issue to harm ecological receptors: soil organisms (microbes, invertebrates, plants), aquatic organisms, wildlife and higher animals, and ecosystem functioning as a whole. This potential was ranked on a scale of 1–5 with guidance notes as in Table S1. A mid-score of score of 2.5 was given for either (a) a likely impact on some organisms in the ecosystem based on toxicological considerations and experience, or (b) existence of plausible mechanisms supporting a possible impact on most organisms in an ecosystem. Factor 6 was allocated a weighting of 2, giving a maximum possible score of 10 (Table 2). This ensured that environmental harms were not understated relative to policy objectives through dilution against other factor scores.

- Factors 7 and 8. Production and trade

These factors were ranked on a scale of 0–5 (Table 2) and are relevant to policy goals supporting sustainability of the regional economy. Trade was considered to be a separate factor to production because several contamination issues of negligible impact on production can have a negative impact on international trade. Presence of specific pesticide residues in food exports may cause a ban on importation in recipient countries, through quantified risks, perceived risks, or for political reasons. For trade a score was allocated based on the potential for the existence of the contamination issue to harm trade, irrespective of the mechanism or motive. Scores for animal welfare and production, and trade, were given weightings of 2 and 1 respectively (Table 2), giving a maximum possible score for these two related factors of 15.

- Factor 9. Reduction of land use flexibility

Some contaminants have the capacity to contaminate land to a degree that reduces its inherent range of viable uses, e.g., a contaminated industrial site may have become unsuitable for residential use without expensive site remediation. Metal accumulation in some pastoral farms may reach levels where if the land use changes to horticulture, vegetables grown in the soil would exceed food standards. Loss of land use flexibility contrasts with policy goals. New Zealand legislation [16] states a purpose of safeguarding the life-supporting capacity of land and soil resources including their capacity to sustain economic activity. Subsidiary regional policy seeks to prevent contamination from causing a reduction in land use flexibility. This issue was addressed independently because it was not specifically captured in other factors. Capacity of a contamination issue to restrict future uses (Factor 9) was allocated a score of 0, 2.5 or 5 (Table 2).

### 2.2.3. Scoring and Allowing for Cross-Compartment Spread

Factor scores for each of the 117 contamination issues were allocated by the authors individually and by consultation, based on experience with environmental contamination issues and familiarity with the specific regional context. Recording of scores and calculations were carried out by spread sheet.

The approach outlined above initially generates sustainability scores for environmental contamination issues as they apply within the air, soil, groundwater, freshwaters, and bed sediments environmental compartments (Table S2). These divisions are useful because they broadly match the corresponding structure for research and policy development work areas currently used at the Waikato Regional Council. However, many significant contamination issues spread between environmental compartments, and a discharge that contaminates air, soil and water could be more important than a discharge that reaches only one of these. For these reasons, a single integrated priority list was also developed, in which issues that affected two or more compartments, or were traceable to a single source, were grouped back into their own parent issue. This was done by selecting and combining scores manually for issues involving significant cross-compartmental spread (Table 3).

The following rules were applied:

1.  For facets of an issue relating to the same environmental compartment, we took the score for the most relevant facet or the average score. This avoided artificial inflation of the score of a parent issue that could occur in cases where several sub-components had been individually scored;
2.  When merging facets of a single wider issue relating to the different environmental compartments, we summed the scores. This gave appropriate numerical credit for inter-compartmental spread of the impacts;
3.  Some issues traceable to a single parent source were simply redefined in that way and re-scored, as described for the parent category 'inorganic contaminants phosphate fertilizers' below.

Examples of the application of these rules to give cross-compartment scores are provided in Appendix A.

### 2.3. Addressing Bias

Bias may be introduced through differences in personal understanding of the scope or effects of each issue. Fortunately, in a relative ranking system the raw score for each factor is less important than where it sits in relation to other factor scores. As a check on appropriate placing, scores for each factor were sorted from high to low, and then compared with nearby scores for other issues. This helped to ensure that individual factor scores for each contamination issue were appropriately placed relative to those recorded for similarly scored issues. More broadly, the ranking approach could be anthropocentric (Table 2). We argue that statutes and policies also reflect the same human values. The stated purpose of New Zealand's principal environmental protection legislation [16], 'to promote the sustainable management of natural and physical resources,' is also anthropocentric. A strength of the modeling approach is that it provides a systematic process and auditable record of decision making. Individual factor scores are shorthand account professional judgments that have been made at a specific point in time. Review of factor scores offers an easy way to determine the reasons why one issue has been ranked more highly than another. In addition, scores and factor weightings can readily be revisited, and new issues can be prioritized in a consistent way, as knowledge develops. This contrasts with the common approach where the reasons behind prioritization decisions are unknown or obscure.

## 3. Results and Discussion

This framework was designed to provide an interface between science and policy as a mean to assign limited resources against the most significant risks (Russell & Gruber 1987). An important feature of the framework is to be robust and objective as the ranking of risks can be influenced by subjectivity and reflect social, cultural, and moral acceptability values (Haynes et al., 1993). There are three key

differences that can be made between this ranking model and traditional source-pathway-receptor Environmental Impact Assessments (EIAs):

(1) The framework of this model is all-encompassing, in the sense that it first sets out a comprehensive list of discharge-based source categories within which any specific environmental contamination issue can be allocated. This approach: (a) provides for the holistic treatment of diffuse contamination issues within a geographic area; (b) avoids predetermination about which issues will be considered (traditional approaches can be biased at the outset by only considering contamination issues that have already been recognized as policy or political priorities), and (c) allows for inclusion of both natural and novel contamination issues (such as environmental contamination caused by geothermal arsenic, or allergenic pollens from exotic tree species in urban areas). This framework also sets a basis for rapid scoring and placement of new issues as they arise, and accommodation of cross-compartmental flows.

(2) The model is designed to work at the interface between policy and science. A core objective was to ensure that the methodology could not become bogged down in mathematical modeling, lost in uncertainties, or paralyzed by the qualitative differences that may exist between two or more diffuse contamination issues—or have to necessarily wait for the results of research that may be years in the future (uncertainties of this type instead form part of the identified risk profiles). The ranking approach used in this model is in fact intentionally devoid of more detailed types of quantitative calculations (for example fugacity modeling, within and between compartment flows, toxicological parameters) which characterize conventional approaches. This makes it possible to prioritize all issues for an entire region or even a country within the one framework, revisit any issue or the complete ranking that at any time, and derive individual rankings in a timely manner. In this sense the model should be seen as more of an advanced triaging process, rather than an EIA-based model or Delphi-type assessment (Lee et al., 2019). It allows for differences in our characterization and understanding between issues and does not predetermine which issues will be included based on prior concern or knowledge. Once top issues are identified, research (including EIA) and policy resources can then be more efficiently devoted to those issues. Each factor in the model represents a policy aspect of issue and harm characterization. We would argue that this type of tool—which forces coherent and systematic thinking aligned to pre-selected policy priority factors—is what many regulatory agencies have been missing. Decisions about policy and spending priorities are made every day, but in the absence of a systematic framework many of these are based more on public concern and political visibility of an issue than its risks to human or environmental health and/or sustainability goals.

(3) The framework is specifically designed to return higher scores for diffuse contamination issues which pose the greatest threats to sustainable management of natural and physical resources, as defined in New Zealand's Resource Management Act (1991) under which its regional councils operate. Under that Act the term 'sustainable management' is defined as managing the use, development, and protection of natural and physical resources in a way, or at a rate, which enables people and communities to provide for their social, economic, and cultural well-being and for their health and safety while (a) sustaining the potential of natural and physical resources (excluding minerals) to meet the reasonably foreseeable needs of future generations; and (b) safeguarding the life-supporting capacity of air, water, soil, and ecosystems; and (c) avoiding, remedying, or mitigating any adverse effects of activities on the environment. This sustainability focus, defined in this way, means that the model allows for issues such as the impact of the ongoing contamination issue on the sustainability of productivity, continuance of trade, and potential economic impacts to be considered. This again differs from other ranking models where the priority may more often be based on mitigating or reducing point-in-time risks to human and ecological health, and the longer term sustainability context view may be inadvertently part of the picture but often takes a back seat. This is not to say that other models do not also rank some of the top issues as this one does, but in this model the sustainability focus is quite intentional.

### 3.1. Individual Compartment Rankings

Scores for 117 environmental contamination issues relating to air, soil, groundwater, freshwater and bed sediment are shown in Table S2, and the distribution of scores and their environmental compartments in Figure 1. For the top 10 ranked issues in each compartment, individual scores for each of the nine factors are provided in Table S3. All 117 issues were assessed for cross-compartment effects or a common parent source. The ranked integrated priority list for the 22 issues that show significant cross-compartment spread or a common parent source is summarized in Table 3.

Although it would be possible to provide a literature-supported discussion about each of the 117 issues, the purpose of the ranking model is to reach past this complexity to pick out issues with the most potential to compromise the policy objective of environmental sustainability. For this reason, and to illustrate the operational use of ranking model, the focus of subsequent discussion is on the top three ranked issues in each environmental compartment.

### 3.1.1. Discharges to Air

The top three ranking air-related issues, and their scores relative to the top air issue, were:

1. AA12.2 Regional impact of greenhouse gases generated globally (100%)
2. AA3.1 Hazardous air pollutants (primarily $PM_{10}$) from domestic home heating in winter (58%)
3. AA4.1 Pollens from pastoral grasses and plantation forestry in rural areas (57%) (Table S2)

The highest-ranking air issue represents regional impacts of climate change caused by global anthropogenic greenhouse gas emissions. This achieved high factors scores for regional scale, accumulation, persistence, and environment impact, with moderately high scores for all other factors apart from potential to cause acute harm to human health (Table S3). Potential regional impacts include higher frequencies of severe rainfall events (causing more flooding and soil erosion), droughts, dust storms and wildfires, increased coastal erosion, storm surges and flooding, and increased concentrations of some aeroallergens (due to longer pollen seasons) and air pollutants ($PM_{10}$ and ground-level ozone) [17–19].

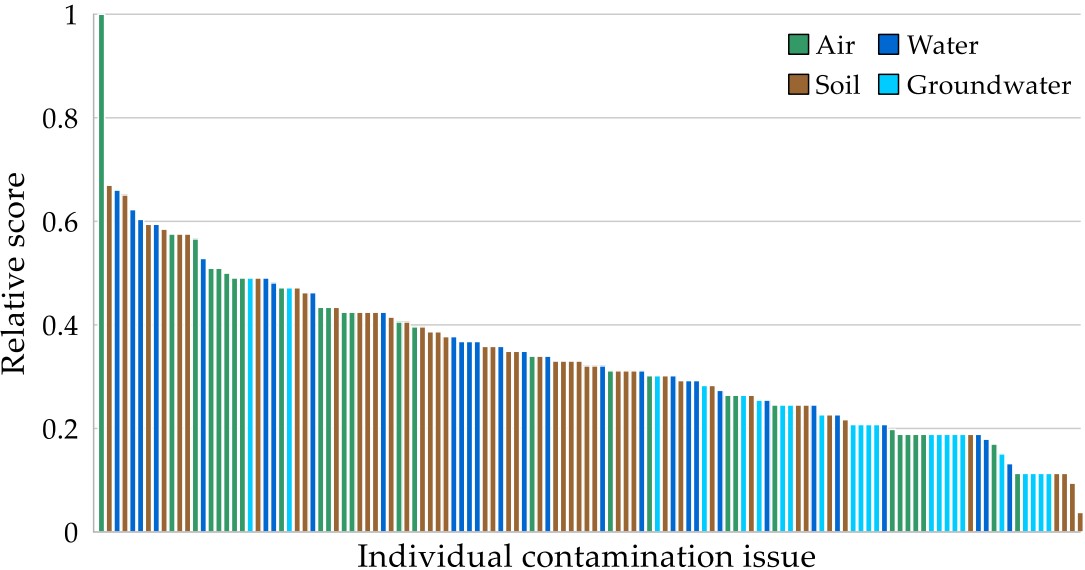

**Figure 1.** Relative scores for 117 individual environmental contamination issues affecting air, soil, groundwater, and aquatic ecosystems. The highest score overall score (53 out of 80, scaled to 1.0) was achieved by issue AA12.2 Regional impact of greenhouse gases generated globally.

**Table 3.** Ranked integrated priority list for 22 issues that show significant cross-compartment effects or a common parent source. All 117 issues were assessed for cross-compartment effects or a common parent source.

| Contamination Issue | Score | Broad Source Category or Categories |
|---|---|---|
| Inorganic contaminants from phosphate fertilizers in soil, groundwater, freshwater and sediment | 90 | Agriculture |
| Excess nutrients (nitrogen and phosphorus) in soils and freshwaters | 86 | Agriculture |
| Global generation of greenhouse gases (impact of) | 86 | Agriculture, energy generation, transport and industry |
| Microbial contamination of freshwater | 57 | Agriculture |
| Zinc accumulation in soil, freshwater, and sediment | 57 | Agriculture |
| Volcanic eruptions | 55 | Natural hazards/ Geothermal |
| Mine tailings sites | 54 | Legacy industrial |
| Altered flow regimes and increased sediment | 54 | Agriculture, forestry, urbanization |
| Arsenic in the Waikato River system | 50 | Natural source with anthropogenic exacerbation (energy generation) |
| Urban air quality | 47 | Domestic home heating, vehicle emissions |
| Steroid hormones from farmed animals | 46 | Agriculture (pastoral) |
| Waikato region generation of greenhouse gases | 43 | Agriculture, energy generation, Transport and industry |
| Natural arsenic and mercury from geothermal areas | 42 | Natural geothermal |
| Exposure to pesticides in rural communities | 41 | Agriculture |
| Assimilable organic carbon in rural waters | 39 | Agricultural |
| Urban storm water runoff | 37 | Urban |
| Use of modern pesticides in pastoral farming | 35 | Agriculture |
| Mercury from coal combustion | 35 | Energy generation |
| Use of treated timber | 34 | Product use |
| Rural use of antibiotics, artificial hormones or other non-pesticide xenobiotic compounds | 33 | Agriculture |
| Landfills | 33 | Industrial and urban |
| Legacy issues from DDT usage | 32 | Agriculture |

Poor urban air quality in winter is the highest ranked regional air issue (30.5 out of 80; Table S2), health impacts of which are primarily attributed to airborne particulates measured as $PM_{10}$ [20]. In most Waikato towns, wood and coal combustion for domestic home heating is the dominant (>80%) winter $PM_{10}$ source [21]. Acute and chronic health impacts of the $PM_{10}$ load range from minor irritation to cardiopulmonary morbidity and mortality [22,23], and societal costs of poor urban air quality in the Waikato region have been estimated at over US\$500 million per annum [20]. Recent research, based on data from the 'Growing up in New Zealand' child cohort study, found that living in a neighborhood with a higher density of wood burners was associated with the increased risk of a non-accidental emergency department visit before the age of three by 28% [24].Two key reasons why this issue achieved a high overall score in this ranking model is that population is treated as an aspect of scale, and issues that have the potential to harm human health are emphasized by weighting (Table 2). Poor urban air quality has become a better characterized problem in New Zealand since the introduction of ambient air monitoring requirements under the Resource Management (National Environmental Standards for Air Quality) Regulations 2004. $PM_{10}$ concentrations have decreased in New Zealand at many locations over the most-recent decade for which complete data is available (2007–16) with improving trends identified in 17 of 39 airsheds (18 of 45 monitoring sites) in winter, when home-heating emissions are at their highest [20,25].

Problems caused by human exposure to pollens ranked third (Table S2). This achieved high scores as a result of their scale and adverse impacts on people, which range from mild inconvenience to severe allergic reactions, anaphylaxis and fatality, particularly among asthmatics [26,27]. Although this category was scored with pollens in mind, a better group term is environmental aeroallergens, because mold spores are also implicated in the same spectrum of adverse health outcomes [27]. Allergy New Zealand provides a seasonal pollen calendar that indicates peak months of pollen release for 22 trees, 5 weed species, 11 grasses, and fungal spores [28]. Some of these plants are native but most are exotic, but for both categories it is evident that most problem pollens are downwind consequences of human activities. Agency responses in New Zealand have extended to designating some plants (e.g., silver birch: *Betula pendula*, privet: *Ligustrum* spp.) as noxious pest species and issuing factsheets, but control measures are limited, e.g., the requirement for a landowner to remove privet relies on receipt of a valid health-related complaint from a neighbor, with the privet causing the problem being within 50 m of the property boundary [29].

### 3.1.2. Discharges to Soil

The top three highest scoring soil resource issues and their scores were:

1. SA2.1 Accumulation of cadmium in rural soils through use of phosphate fertilizers (100%)
2. SA2.2 Accumulation of fluorine in rural soils through use of phosphate fertilizers (97%)
3. SA12.1 Large abandoned mine tailings site (Tui Mine and environs) (88%)

The issues ranked first and second were entered in the model separately but are from the same source—use of phosphate fertilizers on productive land. Approximately 346,000 t of superphosphate fertilizer is applied to Waikato soils annually [12]. Cadmium and fluorine are naturally elevated in phosphate rock used to manufacture these, and their progressive accumulation in pastoral soils has the potential to cause a range of problems discussed elsewhere [30,31] and reflected in factor scores. Both issues score highly for geographic scale, but some factors differ between the two contaminants (Table S3). Increasing cadmium in soil has potential to cause issues with food standard compliance, market access, and human health through increased dietary exposure. Increasing fluorine in soil has the potential to cause chronic fluorosis in grazing animals with loss of productive capacity, and ultimately render large areas of land unsuitable for pastoral farming. Although scored for the Waikato region, these problems will apply globally, because phosphate fertilizers are applied globally [30]. At time of writing, a national strategy to manage the presence of cadmium in New Zealand agriculture is in its ninth year, having recently been reviewed [32,33]. No such strategy exists for fluorine although a subgroup of the Cadmium Management Group has a watching brief as both are derived from the same source.

The issue ranked third is a medium-scale heavy metal contaminated site situated on the western slopes of Mt Te Aroha in the Waikato region, and known as the Tui Mine site. This had been regarded as New Zealand's most contaminated site [34]. The high score came about due to this site's environmental impact, the irreversible nature of contamination coupled with removal of land use flexibility, and potential to cause harm to human health (Table S3). A key environmental impact was acid mine drainage from the mine tailings dam to two streams, which caused extinction of aquatic life for some distance downstream (factor score 5). Geotechnical assessment had also shown that liquefaction of 90,000 m$^3$ of mine waste might occur in a moderate seismic event or following extreme weather, causing failure of the containment dam and landslide that would reach the town below [34]. Since this prioritization system was first developed, the Tui Mine workings and tailings dam have been substantively remediated at a cost of over NZ$22 million [34].

### 3.1.3. Discharges to Groundwater

For groundwater the top three issues were:

1. GA1.1 Discharge of nitrogen to rural groundwater (100%)
2. GA1.4 Discharge of microbial contaminants to rural groundwater (96%)
3. GA1.13 Progressive acidification of rural groundwater (62%)

Scores for the top two of these were very close. The discharge of microbial contaminants to rural groundwater was given high factor scores for scale (regional) and the potential for adverse effects on people or animals that drink contaminated bore-water (Table S3). Livestock waste can carry a variety of bacterial and protozoan pathogens including fecal streptococci, enterococci, campylobacter, human adenovirus, cryptosporidium, giardia and (in the case of poultry) salmonellae [35,36], and discharges into rural surface and groundwater can cause infectious enteric diseases. Daughney and Randall (2009) [37] reported that 23% of groundwater monitoring sites across New Zealand exceeded health-based maximum acceptable value of 1 colony forming unit (cfu)/100 mL. New Zealand's annual incidence of campylobacteriosis is very high compared to the rate in other developed countries [38]. The issue of excess nitrogen in groundwater is similar in both source and scale. High nitrate ($NO_3^-$) in groundwater carries a potential for methaemoglobinaemia or 'blue baby syndrome' in infants younger than 3 months, through gastric bacterial reduction of nitrate to nitrite ($NO_2^-$) [39]. Intensive agriculture has caused an increase in nitrate concentrations in many New Zealand groundwaters.

In this ranking system, discharge of nitrate to groundwater achieved a slightly higher score than microbial discharges, despite the latter achieving a slightly higher score for impact on animal welfare (Factor 7) (Table S3). This was because of a higher score for nitrogen's accumulation in groundwater (Factor 2). Whereas subsoil broadly inhibits the movement and persistence of microbes in groundwater [40–42], nitrate is both persistent in soil and mobile in groundwater. Internationally, research has shown that it can take decades for nitrate discharged into soil to reach groundwater and surface water, meaning that full impacts of current and historic discharges are yet to be felt. This time lag between discharge and ultimate impact of nitrogen filtering through the system has been referred to as the 'nitrate time bomb' [43]. For Factor 2, nitrogen leaching was therefore allocated a higher score (5) than microbial leaching (1) (Table S3).

The third ranking groundwater issue received a significantly lower overall score than the top two issues. Acidification of rural groundwater may come about through nitrification, increased soil carbon dioxide, and use of acidic fertilizers [44–46], countered to an unknown extent by additions of lime. Potential impacts are not well defined, but this issue attracted positive scores for scale, accumulation potential, relative irreversibility, and possible environment impact (Table S3). In the context of this exercise, high rankings for issues that are not yet well characterized signaled that a higher priority should be placed on research.

i. Discharges to aquatic ecosystems

For aquatic ecosystems, the top seven issues were:

1. WA3.1 and WA4.1 Entry of nitrogen, phosphorus and assimilable organic carbon to rural surface waters from direct deposition, runoff or through groundwater (100%)
2. WA2.2 Acidification of oceanic water caused by absorption of global carbon dioxide emissions (94%)
3. WA3.9 Entry of microbial contaminants to rural freshwaters (91%)
4. WA5.1 and WA7.1 Arsenic in Waikato River system—combined impact of natural inputs, Wairakei Power Station discharge and presence of eight hydroelectric dams (90%)
5. WA7.2 Altered flow regimes (equivalent to contamination by pressure and volume) causing destruction and loss of habitat (80%)

6.  WA3.6 Zinc in water and zinc accumulation in rural sediments caused by facial eczema treatments (74%) and

7.  WA1.1 Increased suspended sediment in streams, rivers and lakes as a result of human activity (73%)

Seven issues are listed in this case because issues ranked 1, 3, 5 and 7 comprise a set of inter-related problems caused by intensive agriculture and its land management practices [47–49]. Elevated concentrations of nitrogen, phosphorus and assimilable organic carbon can all result in decreased dissolved oxygen concentrations in small rural streams, either by eutrophication and/or the consumption of oxygen during oxidation of organic matter [50]. Fish and aquatic animals become stressed when dissolved oxygen concentrations fall below 5 mg/L [51]. Agricultural land use practices have a large impact on suspended solid concentrations in streams and rivers in the Waikato River [52]. High suspended solids in aquatic ecosystems can cause reduction of light penetration, impaired visual feeding of fish, loss of macrophytes, smothering of macroinvertebrates, loss of habitats, reduction in species abundance and diversity, and interference with fish migration [53,54]. Waikato regional monitoring has shown that concentrations of nitrogen and phosphorus have increased in many of the Waikato region's rivers over two decades since monitoring began, with an overall pattern for the region being deteriorations (progressive increases) in total nitrogen [55]. In parallel with this, bacterial counts have been too high for safe swimming at nearly 70 per cent of sites sampled, and nearly 75 per cent of cases the water was not suitable for farm animals to drink [56]. Factor scores for each of the four issues ranked 1, 3, 5 and 7 for surface water reflected the scale and impacts presented by each component (Table S3). Although these can also be considered together, this ranking suggests that in priority order the sustainability issues posed to surface waters from intensive agriculture may follow the order increased nutrients > microbial contamination > altered flows > increased suspended sediment.

Regional boundaries include coastal marine areas extending to 12 nautical miles. The second-ranked issue for water (in relation to the Waikato region) is acidification of its coastal oceans caused by increased absorption of carbon dioxide [57], or 'the other $CO_2$ problem' [58,59]. Ocean acidification research is growing exponentially and extensively reviewed elsewhere but this issue projected to impact all areas of the ocean, from the deep sea to coastal estuaries [60]. It is likely to cause widespread, potentially catastrophic, changes to marine ecosystems [61,62]. This issue received high scores for scale, accumulation, lack of reversibility, and environmental impact (Table S2). The resulting high within-compartment ranking complements the high ranking for discharge of greenhouse gases to air (see above).

### 3.2. Comparison with Other Work

Water quality issues in this research are similar to those identified within the US EPA National Water Quality Inventory [63] but the issues are ranked slightly differently due to differences in methodology. The US EPA approach is to review water quality and biological monitoring results to assess the number of miles of waterways which have been impaired. The top ten causes of impairments in US rivers and streams are given as pathogens, habitat alteration, organic enrichment or oxygen depletion of waterways, unknown causes, nutrients, metals, sediment, mercury, flow alteration and turbidity. Excluding oceanic acidification (not considered in the USEPA ranking), there is good agreement about highest priority issues between the US EPA list and this regional-specific prioritization exercise. Further down the regional list, high scores are also achieved for some regionally specific features not seen in the US EPA list. Although these regional features do not score highly at the national scale of the US EPA list or were not considered, they are significant issues at the regional scale. An advantage of our proposed framework is it can be easily adapted for differing scales, as scale does not dominate the scoring system.

These regionally specific features are not discussed further here but include (predominantly natural) geothermal discharges of arsenic to freshwater ecosystems, and agricultural use of zinc as a supplement for the prevention of facial eczema.

Pollutants accumulated from diverse diffuse sources is a water quality management issue. As such, there are other methods that have been developed including the Delphi method to forecast viable solutions to problems where data was missing or incomplete by obtaining the most reliable consensus of a group of experts [64]. This risk-based approach for ranking and prioritizing catchments was used to identify and manage non-point source pollution in Korea [64]. Weaknesses of the Delphi method include reliance on the opinion of the experts and a tendency to eliminate extreme positions and so force a middle-of-the-road consensus. This can be quite time consuming. Our proposed framework requires experts to independently follow a set of rules to score issues rather than come to a consensus. This approach can be quick, allowing emerging issues or new knowledge to be assessed alongside established issues. Extremely divergent positions can be reflected in scores and causes further investigated as part of quality control. In our experience, divergent scores have usually highlighted a knowledge gap within an issue with regulators often taking a more cautious approach compared with academics, e.g., the emerging issue "Entry of microplastics to water" is to be included in the next report update of diffuse contamination issues in the Waikato Region. The relative standard deviation (RSD) for experts scores was 54%, whereas the RSD for established issues is typically <15%.

Decision support systems (DSS) have also been developed to contribute to long-term sustainable development of agriculture practices [65,66]. These DSS are often used to manage specific issues such as nitrate and pesticide pollution from agriculture. However, many issues do not exist in isolation and enduring solutions need to be comprehensive. From experiences in the UK, processes to support decisions and policy to achieve sustainability goals must be holistic and integrate governance approaches that allow for cooperation and joint decision making [8]. Our framework is underpinned by a holistic approach integrating input across sectors. As such, the framework must also include the participation of indigenous people to incorporate knowledge and cultural values. In New Zealand, Maori concepts such as kaitiakitanga (the responsibility to take care of natural resources) and Te Mana o te Wai (restoring and protecting the integrity of water) must be part of decision-making frameworks [67]. Another advantage of our framework is that the integrated list or lists of issues within each environmental compartment can be used at different scales. Applications can be high level, such as governance policy statements or long-term plans, or operational, such as guidance for research focus and spending, and identifying what to monitor both across and within environmental compartments.

Comparison across Compartments

Some issues attained similar or identical scores to each other for a different underlying mix of reasons. In general, air and groundwater-related issues attained lower raw scores than soils or surface water-related issues. Greenhouse gases and urban air pollution are notable exceptions. Outside those issues, it could therefore be argued that overall, some soil and surface water related issues may genuinely be of greater significance in an environmental sustainability context than some air and groundwater-related issues. The focus of the following discussion is on the top five ranked issues within each compartment.

*3.3. Integrated List and Major Themes*

Several contamination issues show substantial spread to two or more environmental compartments, and/or have a common source, and be treated as facets of a single larger issue. Twenty-two cases of this type were identified, and within-compartment scores were aggregated as described in the methodology. Phosphate fertilizers are a source of three significant inorganic contaminants (cadmium, fluorine and uranium), initially as a discharge to soils. Although these contaminants cause different problems, they can be grouped under their common source 'inorganic contaminants in phosphate fertilizers' in an integrated list. To avoid double-counting, the parent source was individually ranked (Table S2) before aggregation of cross-compartment scores. Within compartment scores for 'inorganic contaminants in phosphate fertilizers' were air: 16.5; soil: 50; groundwater: 8, surface waters and bed sediments: 19.5, giving a summed cross-compartment score of 90. The freshwater nutrients nitrogen

and phosphorus were also grouped at this point. The resulting integrated priority list is provided in Table 3. Please note that for both non-aggregated issues and individual components of a newly aggregated issue, the scores remain as provided in Table S2.

The top 22 contamination issues identified within the New Zealand's Waikato region fall within only four key source areas: agriculture (13), urban and industrial (5), volcanic and geothermal (3) and energy generation (3). There is some overlap in the case of some source areas. Greenhouse gases are from agriculture, energy generation, transport, and industry; and arsenic in the Waikato River system starts with a natural source but this has been amplified by use of the geothermal resource and impoundment of the Waikato River for two different types of electricity generation. The key sustainability issues grouped by source category are shown in Figure 2.

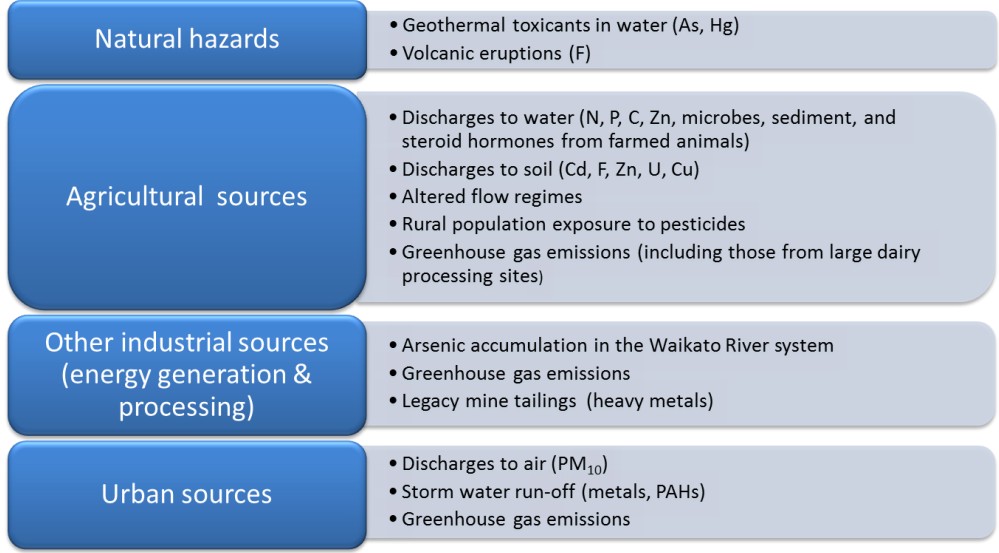

**Figure 2.** Contamination issues in the Waikato region of greatest relevance to environmental sustainability policy goals, and their source categories.

The broad source categories identified in Table 3 and Figure 2 represent a final and optional step applied to the top-ranked issues. Most assignations follow from name and nature of the original identified issue (for example an issue coded with N as the second letter is a natural source and come under the category of natural hazards), but decision making was also supplemented a combination of expert knowledge and the results of over two decades of Waikato Regional Council environmental quality monitoring program data for air, water, soils, and groundwater. By land use area, the region as a whole is dominated by agricultural activities, and contamination issues associated with those (including the proportional contributions made by agriculture compared with other sources) are well characterised. The approach taken in this ranking model differs from others in that it was not presupposed that issues that were already well known to policymakers and/or subject to political awareness were the only potential issues, or would necessarily be highly ranked. That they scored highly was an outcome of the model, rather than a predetermination.

## 4. Conclusions

Environmental contamination issues can be numerous and difficult to characterize and address at a range of levels, and decisions about priorities are often subject to a range of political and other pressures. Although it is routine practice for environmental agencies to set policy objectives, gaps in understanding often exist between how any given contamination issue may relate to each objective. Within the Waikato region, the model shown in this work has so far proven itself to be useful for:

- Enabling the rational comparison of dissimilar issues;

- Identifying which issues pose the greatest challenges to sustainability policy goals;
- Identifying 'hidden' or neglected issues that may pose significant risks to public health
- Delineating and quantifying their key drivers;
- Identifying the main 'problem parent' areas;
- Ensuring that regulatory work-streams are targeted or refocused toward the most significant problems;
- Providing a rationale (where applicable) for not focusing on particular issues where these would detract from work on larger problems;
- Identifying which important issues extend beyond the reach of a regional regulator (e.g., controlling global greenhouse gases); and
- Ensuring efficient use of limited budgetary resources.

The model has its limitations, but its advantages include flexibility, transparency, ability to prioritize new issues as they arise, and ability to identify which issues are comparatively trivial and which present a more serious challenge to sustainability policy goals. This model integrates well as a planning tool and has been used to inform regional policy development.

**Supplementary Materials:** The following are available online at http://www.mdpi.com/2071-1050/12/22/9393/s1, Table S1: Possible factor scores with narrative calibration comments, Table S2: Scores for environmental contamination issues for the Waikato region of New Zealand, Table S3: Individual factors and total scores for the top 10 issues in each environmental compartment.

**Author Contributions:** Conceptualization, N.D.K. and M.D.T.; methodology, N.D.K. and M.D.T.; validation, N.D.K., M.D.T., J.C., A.R., O.C and L.A.T.; formal analysis, N.D.K., M.D.T.; investigation, N.D.K., M.D.T., J.C., A.R., O.C. and L.A.T.; writing—original draft preparation, N.D.K., M.D.T., and L.A.T.; writing—review and editing, N.D.K., M.D.T., J.C., O.C., and L.A.T.; funding acquisition, N.D.K., M.D.T. and L.A.T. All authors have read and agreed to the published version of the manuscript.

**Funding:** This research was funded by the Diffuse Contamination Programme of the Waikato Regional Council with additional funding from the New Zealand Ministry of Business, Innovation and Employment (MBIE), grant number CAWX1708.

**Conflicts of Interest:** The authors declare no conflict of interest. The funders had no role in the design of the study; in the collection, analyses, or interpretation of data; in the writing of the manuscript, or in the decision to publish the results.

## Appendix A

Example 1: Greenhouse gas issues.

The first two rules allowing for cross-compartment spread are illustrated with the four greenhouse gas discharges and their raw scores (Table A1).

**Table A1.** Scores for all greenhouse gas related issues, with adjusted scores denoting those carried forward for ranking Waikato impacts of global greenhouse gas emissions.

| Compartment | Issue | Raw Score (Out of 80) | Adjusted Score |
|---|---|---|---|
| *Air* | AA12.2 Global generation of greenhouse gases | 53 | 53 |
| *Air* | AA12.1 Greenhouse gases generated within the Waikato Region | 26 | 0 |
| *Water/sediments* | WA2.2 Acidification of oceanic water caused by absorption of global carbon dioxide emissions | 33 | 33 |
| *Water/sediments* | WA2.3 Acidification of oceanic water caused by Waikato carbon dioxide emissions | 16.5 | 0 |
| **Sum across compartments:** | | | **86** |

The discharges to air, AA12.1 Greenhouse gases generated within the Waikato Region are a subset of AA12.2 Global generation of greenhouse gases. Any number of additional subsets, such as Greenhouse gases generated in the town of Hamilton, could also be generated, and summing these would artificially inflate the overall parent issue. In this case, the scores should not be averaged or summed, but kept separate, and which one we use depends on whether we are interested in global or Waikato-only greenhouse gas emissions. The same applies to the Acidification of oceanic water caused by global carbon dioxide emissions, and the Waikato subset. The total score for the one parent issue (issues associated with global greenhouse gases) is the sum of the two "global" scores for air and water, because the effects of the discharge are felt over two major environmental compartments: air and water. The sum is 53 + 33 = 86. If the focus is on issues associated with Waikato region generation of greenhouse gases, the alternative pair are added: 26 + 16.5 = 42.5. The overall score for the parent issues are therefore:

- Waikato-related issues associated with the global generation of greenhouse gases: 86
- Waikato-related issues associated with greenhouse gases generated in the region: 42.5

Example 2. Trace elements in phosphate fertilizers.

Some judgment is required in deciding which issues can be safely aggregated. An example is whether to treat inorganic contaminants in phosphate fertilizers as one issue (and thus average the scores that are above 40% for cadmium, fluorine and uranium within the soil compartment), or whether to retain a separation based on each contaminant, in which case taking the sum of scores for each contaminant would be more appropriate. In this work a decision was made to maintain a distinction between cadmium, fluorine and uranium on the basis that known of potential effects differ between these three elements. However, in merging issues for each of these three contaminants, it is still necessary to account for known or potential impacts across compartments. This is illustrated for scores for issues relating to cadmium in phosphate fertilizers in Table A2.

**Table A2.** Scores and adjusted scores for all cadmium related issues.

| Compartment | Issue | Raw Score (Out of 80) | Adjusted Score |
|---|---|---|---|
| Soil | SA2.1 Accumulation of cadmium in rural soils through use of phosphate fertilizers | 35.5 | 35.5 |
| Food | FA1.2 Cadmium in grains and vegetables | 30 | 0 |
| Groundwater | GA1.8 Potential discharge of cadmium to rural groundwater | 6 | 6 |
| Water/sediments | WA3.2 Trace element contaminants in phosphate fertilizers that enter surface waters | 19.5 | 6.5 |
| Air | AA5.1 Dispersal of phosphate dust and PM10 to air from superphosphate application in rural areas | 16.5 | 0 |
| **Sum across compartments:** | | | **48** |

In this case, the score for food is discounted (rule 3 is applied) because accumulation of cadmium in food is already ranked as part of the score for SA2.1 Accumulation of cadmium in rural soils. To include food would double-count this impact. The second adjustment made is that the score of WA3.2 Trace element contaminants in phosphate fertilizers that enter surface waters is divided by three. This issue was intended to cover all three contaminants (cadmium, uranium and fluorine), so needs to be factored down in order to be used as part of a cadmium aggregate. The third adjustment is that the score for AA5.1 Dispersal of phosphate dust and PM$_{10}$ to air from superphosphate application in rural areas air is also set to zero. In this case, the reason is that the primary impact would be related to particulates themselves, and/or the presence of uranium, which introduces potential for exposure of lung tissue to radon-222 and its daughter products from any fine particles that lodge in the lungs. In other words, this is not a cadmium issue. Fluorine and hydrogen fluoride in phosphate

dust would also bring the potential for an acute respiratory irritation, which may also contribute to cytokine-mediated inflammation as seen with $PM_{10}$.

Once these adjustments are made, the scores can be summed across compartments (soil, groundwater, and water/sediment) to yield a total score for cadmium in phosphate fertilizers of 48.

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
