# Peer review of "Development and Deployment of a Framework to Prioritize Environmental Contamination Issues"

_sustainability, doi:10.3390/su12229393_

Round 1

Reviewer 1 Report

The study shows a way to consider environmental contaminants in a rural context (case of New Zealand).

The environmental compartments are well described but the ranking approach ($ 2.2.2) is not well explained. 

Could you give more details about the scoring scale, your choice of factor weighting and the highest possible score ? You must detail your scoring scale : 

what means a score of 1 ? 2? 3? .... for your 9  factor model in Table 2?

The table S1 (to which I have no access for my review) must be in the main text to improve the understanding and interest of the reader.

You can explain it in a synthetic table for example.

Explain in detail also your choice of factor weighting for the nine factors.

In particular, why the scoring scale isn't always the same to uniformize your method to class your data ? If the Factor weighting is the only parameter to vary, you will have a more comprehensible evaluation of the impact of the nine factors  choosen to evaluate environmental contamination parameters.

In your conclusions, could you more explain the limitations and advantages of your model (lines 557 to 560) ?. 

Author Response

Reviewer 1

The study shows a way to consider environmental contaminants in a rural context (case of New Zealand).

The environmental compartments are well described but the ranking approach ($ 2.2.2) is not well explained. 

Reply: We thank the reviewer for the relevant and constructive comments. We received these comments after Reviewers 2 and 3 and many comments have been addressed. The ranking approach has been further clarified with additional text in both Sections 2.2.1 and 2.2.2 as described below for Reviewer 2’s comment 1. 

Could you give more details about the scoring scale, your choice of factor weighting and the highest possible score ? You must detail your scoring scale : 

Reply: Reviewer 2 also had a similar comment and we have provided extensive clarification at the beginning of Section 3 to better define how the scoring was used in our framework.

what means a score of 1 ? 2? 3? .... for your 9  factor model in Table 2?

Reply: There was a suggestion that we find very useful to explain how the framework was used and we provided examples as Appendix before the References section.

The table S1 (to which I have no access for my review) must be in the main text to improve the understanding and interest of the reader.

You can explain it in a synthetic table for example.

Reply: Again, we had a similar query from Reviewer 2 and we will make sure that the Sup Material will be available. We decided to include Table S1 in Sup Mat as it is quite extensive, and we wanted to keep the manuscript to a workable size. It is already long and we feel that Table S1 can be accessed by experts that want to use the framework but does not affect the effectiveness of the manuscript.

Explain in detail also your choice of factor weighting for the nine factors.

In particular, why the scoring scale isn't always the same to uniformize your method to class your data ? If the Factor weighting is the only parameter to vary, you will have a more comprehensible evaluation of the impact of the nine factors  choosen to evaluate environmental contamination parameters.

Reply: The Factors are extensively explained in section 2.2.2 and we have amended to further clarify. The inclusion of examples in Appendix provides further details on how these factors have been applied and can be modified to incorporate other criteria.

In your conclusions, could you more explain the limitations and advantages of your model (lines 557 to 560) ?. 

Reply: Thank you for this relevant comment that has been identified by the other reviewers. We have amended section 3.2 by adding more details about weaknesses and advantages of our Framework. We now feel that these provide sufficient information describing the pros and cons of this framework and adaptability to other contexts.

Reviewer 2 Report

The paper "Validation of a framework to prioritise environmental contamination issues" faces with a very relevant issue. It is proposing a way to define priority in environmental issues (specifically, the case of environmental contamination) for regulatory agencies.

The article acknowledges the difficulty of defining such a complex framework and specifies that the model refers to a specific case (in New Zealand and a rural context). The method, however, should be flexible and therefore usable in other contexts, considering other specific factors and variables. Although the paper is a high-quality article, it needs some clarification.

In short:

1. It is not clear how it defined the list in "Table 1" (literature review? Expert interviews? Previous research?). Even if the list in the "Table 1" is an outcome of authors' considerations, readers need to know it and to understand better why and how authors consider that list a good approximation to the reality; they need further information.

2. For "Table 2" it is not clear whether it is the product of comparison with experts and if a specific method was used. The "Factor weighting" or the "Scoring scale" appear challenging to understand. Also, it is not possible to access some tables (e.g., "Table S1" or "Table S2") that seem to help clarify this aspect, but at the moment, reviewer(s) can not access to them.

3. On page 8 of the pdf file, it would be useful to propose some examples of adopted rules, maybe even inserting a real case of calculation adopted in an Annex.

4. The connection between "Table 3" and "Figure 2" is fundamental to define not only the policy priorities but also the source category. It is not clear, however, in what way the "significant cross-compartment or a common parent source" has been defined. Is this also the result of an expert consultation? Or is it the product of the researchers' evaluations? On what basis?

Considering the paper purposes, it seems useful to clarify the issues listed above.

Author Response

Please see the attachment for Reviewer 2

Reviewer 3 Report

This paper would initially appear to present a novel approach to identifying and prioritising environmental issues for policy development and management decision making. It is well written and comprehensive in the area it covers but seems to be basically a risk assessment process, so not particularly novel (see the following for examples of risk assessment in prioritisation and policy framing - Asante-Duah, D K. (1998)  Risk assessment in environmental management. United States: Dept of Energy; Linkov, I. (Ed.). (2004). Comparative risk assessment and environmental decision making (Vol. 38). Springer Science & Business Media; Russell, M., & Gruber, M. (1987). Risk assessment in environmental policy-making. Science, 236(4799), 286-290; Haynes, K. E., Ratick, S., Bowen, W. M., & Cummings-Saxton, J. (1993). Environmental decision models: US experience and a new approach to pollution management. Environment International, 19(3), 261-275; Hammitt, J. K. (2019). Probability is All We Have: Uncertainties, Delays, and Environmental Policy Making (Vol. 10). Routledge).

That said, the context (holistic assessment of multiple connected environmental regions and impact sources) is quite interesting but the paper would just seem to be a little confused as to purpose. The actual description of the impact issues in the Waikatoto region is in itself quite interesting but not what I thought the paper was about and perhaps less novel. What may be novel is the understanding of why and how the criteria were determined and the implications of this for the relative rankings identified, but this process is not described in detail.

This would seem to be an expert elicitation/ judgement exercise (cf Delphi analysis) although this aspect of the study is not well described and it would be good to better understand how that expert judgement was applied – that is the detail that enables a decisions as to how easily transferrable or broadly applicable it might be. The authors explain in detail the criteria included in the components of the assessment but not how those criteria were derived and the weightings/ scores actually assigned, this would be useful context. Not entirely sure why the authors haven’t compared their approach to standard risk assessments or other approaches and identified the relative benefits – this would be very useful. They do allude to such approaches but not as a direct comparison.

I was a little confused through the discussion as to the purpose of the paper, is it to introduce the approach for identifying risk and prioritising, or is it about the risks themselves in the regional context? The discussion seemed to focus on the latter whilst the introduction and materials and methods seemed to be more about the approach and relevance for decision making. The discussion dealt more with the implications and impacts of the top ranked issues in the NZ context than the process of ranking and implications of the approach for policy. In fact the comparison with other work section focused on the output of the ranking and how the prioritisation compared with elsewhere cf the approach itself? Whilst the identification of issues and comparison of causes is quite interesting in itself I think the authors need to be mindful of the objective implied in the title “Validation of a framework to prioritise environmental contamination issues” and return to this point regularly throughout the discussion.

Specific Comments

L56-57 Typo – source of anthropogenic contaminants

L74 outcomes of results cf just outcomes

L1101-114 This sounds like expert elicitation, how many experts were involved in establishing these relationships.

L127-128 Seems to be term/ text missing here

L169-161 Not clear what is meant by pooling, need to clarify.

L162 need to explain the particulate matter abbreviations on first introduction and the scaling levels.

L179 – text needs review for English

L175 Is the context here not actually Irreversibility as the highest scores are associated with persistence?

L240-243 Would be useful here to establish i) the expertise of the authorship team and ii) the nature and extent of the consultation.

L244 are these sustainability scores given the measure/ assessment is largely of impact?

L246-247 this point needs clarification – what research and policy development work and how?

L252 in a standard risk assessment process multiple interaction pathways is often a feature of the analysis. Need to clarify the selection and combination process.

L301-302 This legend needs clarification as it would seem to infer it is a subset of scores and a secondary selection process, but aren’t all the scores assessed for cross compartment or common parent source and this is in fact just those scores highest on the list? If so then maybe worth including cut-off point and why, and what proportion of the total issues this subset represents.

L475-478 provides a good example of where the authors could use the findings to discuss the key point of the paper and validate the framework by clarifying why using this approach “high scores are also achieved for some 476 regionally specific features not seen in the US EPA list”.

L479-492 comes to the crux of the paper and alludes to why risk assessment is important and the different approaches possible, but unfortunately the authors don’t actually explain the reasons why the proposed approach is better? This is important and the point where the authors could clearly emphasise the uniqueness and novelty of their work or even why they have chosen to take this different approach.

L493-503 This seems at odds with what the paper actually presented? The statement that “This article aims to review the ‘state of the art’ in EPI research and practice from the perspective of its conceptual meaning, processes of implementation and outcomes ‘on the ground’.” Is inconsistent with the current content.

Author Response

Please see the attached file below for Reviewer 3

Round 2

Reviewer 1 Report

The comments of the reviewers were well taken into account by the authors.

The answers are good, and useful supplements have been added to the text increasing the interest of the reader.

The article is interesting and deserves to be published in the journal Sustainability.

Reviewer 3 Report

The authors have comprehensively addressed my comments and have modified the manuscript accordingly. I must applaud them for doing such a thorough job.

I like the authors new distinction of this framework as “an advanced triage tool” that along with the additional explanatory text I believe makes it much clearer what the tool is and how it aligns with other risk assessment approaches.

They have also made the key differences between this ranking model and traditional source-pathway-receptor Environmental Impact Assessments much clearer.

They have clarified and improved the description of how the criteria underpinning the tool were determined and how this relates to the specific management structures, and therefore makes the tool most useful.

They have clarified the expert elicitation process so that it now much clearer how the scores were derived.

They have also clarified the key differences with other possible approaches and the benefits of this one for the particular application.

I support the change in the title and believe that this will definitely help avoid confusion.

I all I think they have done an excellent job in revising the manuscript and as such I would be happy to see this paper published.